# Transcriptomic and Metabolic Profiling of High-Temperature Treated Storage Roots Reveals the Mechanism of Saccharification in Sweetpotato (*Ipomoea batatas* (L.) Lam.)

**DOI:** 10.3390/ijms22136641

**Published:** 2021-06-22

**Authors:** Chen Li, Meng Kou, Mohamed Hamed Arisha, Wei Tang, Meng Ma, Hui Yan, Xin Wang, Xiaoxiao Wang, Yungang Zhang, Yaju Liu, Runfei Gao, Qiang Li

**Affiliations:** 1School of Life Science, Jiangsu Normal University, Xuzhou 221116, China; 1020180010@jsnu.edu.cn (C.L.); tangweilhr@gmail.com (W.T.); 2020190639@jsnu.edu.cn (M.M.); yanhuijsxz@jaas.ac.cn (H.Y.); 2020180466@jsnu.edu.cn (X.W.); 2Xuzhou Institute of Agricultural Sciences in Jiangsu Xuhuai District/Key Laboratory of Biology and Genetic Breeding of Sweetpotato, Ministry of Agriculture and Rural Affairs, Xuzhou 221131, China; 20081003@jaas.ac.cn (M.K.); wangxin@jaas.ac.cn (X.W.); zhangyungang@jaas.ac.cn (Y.Z.); yajuliu@jaas.ac.cn (Y.L.); grf15852141027@gmail.com (R.G.); 3Department of Horticulture, Faculty of Agriculture, Zagazig University, Zagazig, Sharkia 44511, Egypt; mohhamedarisha@gmail.com

**Keywords:** sweetpotato, sweetness, glycometabolism, saccharification, transcriptome, metabolome

## Abstract

The saccharification of sweetpotato storage roots is a common phenomenon in the cooking process, which determines the edible quality of table use sweetpotato. In the present study, two high saccharified sweetpotato cultivars (Y25, Z13) and one low saccharified cultivar (X27) in two growth periods (S1, S2) were selected as materials to reveal the molecular mechanism of sweetpotato saccharification treated at high temperature by transcriptome sequencing and non-targeted metabolome determination. The results showed that the comprehensive taste score, sweetness, maltose content and starch change of X27 after steaming were significantly lower than those of Y25 and Z13. Through transcriptome sequencing analysis, 1918 and 1520 differentially expressed genes were obtained in the two periods of S1 and S2, respectively. Some saccharification-related transcription factors including MYB families, WRKY families, bHLH families and inhibitors were screened. Metabolic analysis showed that 162 differentially abundant metabolites related to carbohydrate metabolism were significantly enriched in starch and sucrose capitalization pathways. The correlation analysis between transcriptome and metabolome confirmed that the starch and sucrose metabolic pathways were significantly co-annotated, indicating that it is a vitally important metabolic pathway in the process of sweetpotato saccharification. The data obtained in this study can provide valuable resources for follow-up research on sweetpotato saccharification and will provide new insights and theoretical basis for table use sweetpotato breeding in the future.

## 1. Introduction

Sweetpotato is widely cultivated in China and is used as an important food source, feed material, and industrial raw material because it is rich in carbohydrates, nutrients, and functional ingredients [1]. The carbohydrate composition of sweetpotato mainly includes starch, soluble sugar (fructose, sucrose, glucose, maltose), cellulose, etc. [2]. The different carbohydrate composition determines the quality and use of sweetpotato cultivars [3]. The breeding objectives of different sweetpotato types, including table use type, starch type and functional type, are to obtain the optimal combination of these nutrient components and contents.

Sweetpotato breeding for table use is one of the main objectives in China. For table use sweetpotato cultivars, the sweetness due to storage root soluble sugar content during cooking is a vital indicator to evaluate the quality of the cultivar. In fresh sweetpotato storage roots, soluble sugar mainly includes glucose, sucrose, and fructose, while maltose is produced in large quantities after cooking, resulting in high soluble sugar content [4,5,6]. The soluble sugar content also affected the moistness, taste, and adhesion of storage roots after cooking [7,8]. It was found that the maltose concentration was significantly, positively correlated with sweetpotato sweetness [9]. Hence, maltose is considered the most important source of sweetpotato sweetness.

Saccharification refers to the process in which tissues rich in starch or cellulose undergo acidolysis or hydrolysis to produce sweet substances in the presence of water [10]. Saccharification of sweetpotato storage roots is a process in which the starch is converted into soluble sugar during cooking. Maltose is the final product of sweetpotato saccharification, which determines the final sweetness [11,12,13]. During the saccharification of sweetpotato, the synthesis of maltose is influenced by many factors, including the activity of amylase and the characteristics of starch structure. Sweetpotato cultivars (i.e., Satsumahikari and Okikogane) have extremely low sweetness, almost no maltose production, and extremely low β-amylase activity after cooking [9]. β-amylase is a thermostable enzyme that hydrolyses starch during cooking of sweetpotato storage roots, thereby influencing eating quality. Some sweetpotato genotypes had little or no β-amylase activity in their storage roots. The sweetness in these genotypes cannot increase during cooking because there is insufficient β-amylase involved in the normal hydrolysis of starch to produce maltose [13]. The maltose content per fresh weight of steamed roots increased to about 10% (*w*/*w*) with β-amylase activity up to about 0.2 mM maltose/min/mg protein, and maltose content was maintained at that level with activity up to 0.55 mM maltose/min/mg protein [9]. β-amylase is directly involved in maltose synthesis, but there are still other factors that regulate maltose production.

Starch gelatinization provides the necessary substrate for β-amylase because it is unable to decompose raw starch granules [14]. With the increase of the starch gelatinization temperature of sweetpotato and decrease of maltose concentration, there was a significant negative correlation between the maltose concentration and starch gelatinization temperature of sweetpotato; however, the negative correlation was more obvious when β-amylase activity was enhanced [9,15]. Starch gelatinization temperature of sweetpotato was closely correlated with the molecular structure of amylopectin including the chain length and arrangement type. The short chain (DP6-10) of amylopectin was significantly negatively correlated with gelatinization temperature, while the long chain (DP12-17) was significantly positively correlated with gelatinization temperature [16]. At the same time, the structure of amylopectin in sweetpotato starch was affected by soil temperature and growth conditions [17]. The growth environment of sweetpotato regulates the activity of amylopectin synthesis and modification-related enzymes which determine the structural characteristics of amylopectin, and ultimately affect the gelatinization temperature of starch and indirectly regulates the synthesis of maltose during sweetpotato saccharification. In a certain sense, the genes involved in the synthesis and modification of amylopectin can influence the saccharification of sweetpotato, especially SS (starch synthase) and BE (branching enzyme) [18]. However, the specific gene regulatory network related to sweetpotato saccharification and maltose synthesis is not clear.

Research on sweetpotato saccharification is mainly focused on the physiological and biochemical aspects, and is rarely related to molecular and metabolic aspects. In this study, three sweetpotato cultivars with extreme saccharification were used to analyze their differential genes and metabolites under high temperature by transcriptome and metabolite. The purpose of this study is to reveal the molecular mechanism of saccharification of sweetpotato storage roots and to provide a reliable theoretical basis for the breeding of table use sweetpotato.

## 2. Materials and Methods

### 2.1. Plant Materials and Treatments

Three sweetpotato cultivars were used for the current experiments, including Y25/Y (*Ipomoea batatas* (L.) Lam. cv ‘Yanshu 25’), Z13/Z (*Ipomoea batatas* (L.) Lam. cv ‘Zheshu 13’), and X27/X (*Ipomoea batatas* (L.) Lam. cv ‘Xushu27’). Y25 contains a light red skin, orange flesh storage roots and Z13 has red skin and yellow flesh storage roots with a high-saccharification rate. X27 has a red skin and white flesh storage roots with a low-saccharification rate. These 3 cultivars were obtained from Xuzhou Institute of Agricultural Sciences in Jiangsu Xuhai District, Zhejiang Academy of Agricultural Sciences and Yantai Academy of Agricultural Sciences, respectively (Figure 1).

The 3 cultivars were planted in the modern agricultural experimental farm of Xuzhou Institute of Agricultural Sciences on 30 May 2019 (coordinates: 117.407 E, 34.293 N). Three sweetpotato cultivars were randomly arranged in blocks, and each cultivar was planted in 5 rows, of which 10 plants were planted in each row. The single ridge planting mode was adopted, and the plant spacing was set to 25 cm and the row spacing was set to 85 cm. The samples were collected at 90 days after transplanting (early harvest, S1) and 130 days (normal harvest, S2). Ten plants of each cultivar were randomly harvested and 3 repetitions were set for a total of 30 plants. The storage roots with medium size, complete appearance, and smooth skin were screened for chemical determination.

After harvest, the fresh sweetpotato were washed and the surface dried with absorbent paper. Sweetpotatoes were randomly divided into two groups including contrast check samples (CK) and treated samples (T) and 5 pieces for each group were taken. CK samples were directly cut into 0.5 cm × 0.5 cm × 0.5 cm cubes and immersed in liquid nitrogen for 30 min. A full storage root of treated samples was placed in a fresh-keeping bag (to prevent moisture loss) and heat-treated in a constant temperature incubator at 60 °C for 90 min. Then treated samples were quickly cut into 0.5 cm × 0.5 cm × 0.5 cm pieces and immersed in liquid nitrogen for 30 min. All the samples were stored at –80 °C.

### 2.2. Measurements of Glucose, Fructose, Sucrose, and Maltose

The sugar components of sweetpotato storage roots with different treatments (CK, 60 °C, 80 °C, 90 °C, and steaming) were determined by HPLC (high performance liquid chromatograph) which was modified with reference to Chan [19]. Then, 0.3 mol/L potassium ferrocyanide and 1.0 mol/L zinc acetate were used as precipitant in sample treatment. ELSD (evaporative scatter light detector) and carbohydrate analysis column (model: 4.6 mm × 250 mm, Agilent Technology Co. LTD, Santa Clara, CA, USA) were used in the experiment. The mobile phase was 70% acetonitrile and the flow rate was 1 mL/min. The drift tube temperature was set to 80 °C. Four sugar standards including glucose, fructose, sucrose, and maltose (Sigma Chemical Co., St. Louis, MO, USA) were selected as internal controls in the experiment. Three independent repetitions were carried out for each sample.

### 2.3. Measurements of Starch, Amylose, Total Amylase Activity, and Starch Gelatinization Temperature

Starch determination kit (Code: YX-W-C400, Lai-Er Biotechnology Co., LTD, Anhui, Chain) was used to assay the starch content of sweetpotato storage roots. Amylose content was determined by Amylose/amylose Detection Kit (Code: K-AMYL, Megazyme, Ireland). The total amylase activity of storage root samples of 3 sweetpotato cultivars treated at different temperatures (CK, 60 °C, 80 °C, 90 °C, and steaming) was determined by plant amylase (AMS) ELISA Kit (Code: CK-E91026, Ruixin Biotechnology Co., Ltd., Guangxi, China). The determination of the above indexes was carried out strictly according to the operation instructions of the kit.

Starch gelatinization temperature was determined by rapid viscosity measurement (RVA). A weight of 3.0 g sweetpotato starch which was obtained by water extraction and vacuum freeze drying was placed in RVA gelatinization box then 25 mL ddH_2_O were added, stirred, and evenly put into the instrument. The RVA rotor was rotated at 960 rpm/min speed for 10 s, and then kept at 160 rpm/min speed until the end of the experiment. The initial temperature was kept at 50 °C for 1 min, then it was raised to 95 °C and kept for 4.5 min, and then decreased to 50 °C for 4.0 min and maintained for 3 min. The whole process continued for 16.7 min. The data was analyzed by RVA data analysis software.

### 2.4. Taste Evaluation of Sweetpotato

Six trained evaluators scored the sweetness (0–5), viscosity (0–5), and comprehensive taste score (0–5) of cooked sweetpotato storage roots, and carried out statistical analysis.

### 2.5. RNA Isolation cDNA Library Preparation and RNA-Seq

Total RNA was extracted from 3 sweetpotato cultivars roots treated at 60 °C and CK using Trizol reagent kit (Invitrogen, Carlsbad, CA, USA) according to the manufacturer’s protocol. RNA quality was assessed on an Agilent 2100 Bioanalyzer (Agilent Technologies, Palo Alto, CA, USA) and checked using RNase free agarose gel electrophoresis. After total RNA was extracted, eukaryotic mRNA was enriched by Oligo (dT) beads, while prokaryotic mRNA was enriched by removing rRNA by Ribo-ZeroTM Magnetic Kit (Epicentre, Madison, WI, USA). Then the enriched mRNA was fragmented into short fragments using fragmentation buffer and reverse transcripted into cDNA with random primers. Second-strand cDNA were synthesized by DNA polymerase I, RNase H, dNTP, and buffer. Then the cDNA fragments were purified with QiaQuick PCR extraction kit (Qiagen, Venlo, The Netherlands), end repaired, poly(A) added, and ligated to Illumina sequencing adapters. The ligation products were size selected by agarose gel electrophoresis, PCR amplified, and sequenced using Illumina HiSeq2500 by Gene Denovo Biotechnology Co. (Guangzhou, China).

### 2.6. Analysis of Differentially Expressed Genes (DEGs)

RNAs differential expression analysis was performed by DESeq2 software [20] between two different groups (and by edgeR [21] between two samples). The genes/transcripts with the parameter of false discovery rate (FDR) ≤ 0.05 and absolute fold change ≥ 2 were considered as differentially expressed genes/transcripts [22].

### 2.7. Gene Ontology (GO) and Kyoto Encyclopedia of Genes and Genomes (KEGG) Enrichment Analysis

GO [23] and KEGG [24] pathways of these DEGs were analyzed to determine the significant abundance of GO terms and metabolic pathways. GOSeqR software package was used for GO enrichment analysis, and KEGG Orthology software was used for KEGG pathway analysis. Functional terms and pathways of DEGs were considered to be significantly different when *p* < 0.05. The WEGO software was used to classify the functional classification of the GO annotations.

### 2.8. Real-Time Quantitative PCR (qRT-PCR) Validation

Nine randomly selected unigenes were used for quantitative RT-PCR assays using QuantStudio ^TM^ 6 Flex Real-Time PCR System by Thermo Fisher Scientific. These unigenes included G25158, G15055, G6471, G18261, G28456, G35317, G18147, G39183, and G11136. Total RNA was extracted from sweetpotato storage roots by Total RNA rapid extraction kit (GK3016, Shanghai Generay Biotech Co., Ltd., Shanghai, Chain) and was reverse-transcribed into cDNA using ReverTra Ace^®^ qPCR RT Master Mix with gDNA Remover Kit (FSQ-301, TOYOBO Co.,Ltd.,Japan). The qRT-PCR reaction protocol was conducted using SYBR Green Real Time PCR Master Mix (10 μL), PCR Forward/Reverse Primer (10 μM, 0.5μL), cDNA template (1 μL), ddH_2_O (8 μL), and ARF (JX177359) was used as the internal control (reference gene) [25]. qRT-PCR primers of each unigene were designed using primer 3 Plus software are shown in Appendix A. The data from 3 independent repetitions were calculated with the 2^−^^△△Ct^ method [26].

### 2.9. Metabolites Extraction and Identification

Next, 50 mg of the sample were weighted in Eppendorf tubes. After the addition of 1000 μL of extraction solvent (acetonitrile: methanol: water, 2:2:1, containing internal standard), the samples were vortexed for 30 s, homogenized at 45 Hz for 4 min, and sonicated for 5 min in an ice-water bath. The homogenate and sonicate circle was repeated 3 times, followed by incubation at –20 °C for 1 h and centrifugation at 12,000 rpm/min and 4 °C for 15 min. The resulting supernatants were transferred to LC-MS vials and stored at –80 °C until the UHPLC-QE Orbitrap/MS analysis. The quality control (QC) sample was prepared by mixing an equal aliquot of the supernatants from all the samples.

LC-MS/MS analyses were performed using a UHPLC system (1290, Agilent Technologies) with a UPLC HSS T3 column (2.1 mm × 100 mm, 1.8 μm) coupled to Q Exactive (Orbitrap MS, Thermo). The mobile phase A was 0.1% formic acid in water for positive, and 5 mM/L ammonium acetate in water for negative, and the mobile phase B was acetonitrile. The elution gradient was set as follows: 0 min, 1% B; 1 min, 1% B; 8 min, 99% B; 10 min, 99% B; 10.1 min, 1% B; 12 min, 1% B. The flow rate was 0.5 mL/min. The injection volume was 2 μL. The QE mass spectrometer was used for its ability to acquire MS/MS spectra on an information-dependent basis (IDA) during an LC/MS experiment. In this mode, the acquisition software (Xcalibur 4.0.27, Thermo Electron Co., Waltham, Massachusetts, USA) continuously evaluates the full scan survey MS data as it collects and triggers the acquisition of MS/MS spectra depending on preselected criteria. ESI source conditions were set as following: Sheath gas flow rate as 45 Arb, Aux gas flow rate as 15 Arb, Capillary temperature as 320 °C, Full ms resolution as 70,000, MS/MS resolution as 17,500, Collision energy as 20/40/60 eV in NCE model, and spray voltage as 3.8 kV (positive) or –3.1 kV (negative), respectively.

### 2.10. Qualitative and Quantitative Analysis of Metabolites

MS raw data files were converted to the mzML format using ProteoWizard and processed by R package XCMS [27] (version 3.2), including retention time alignment, peak detection, and peak matching. Then the data were filtered by the following criterion: The metabolite would be retained when it was detected in more than 50% samples (including QC sample). Normalization to an internal standard [28] for each sample was done subsequently. Next, missing values were replaced by the half of the minimum value found in the dataset by default [29]. The preprocessing results generated a data matrix that consisted of the retention time (RT), mass-to-charge ratio (m/z) values, and peak intensity. OSI-SMMS (version 1.0, Dalian Chem Data Solution Information Technology Co. Ltd., Dalian, Liaoning, China) was used for peak annotation after data processing with in-house MS/MS database. The metabolic data of different comparison groups were analyzed by constructing an orthogonal partial least squares model using the R package ropls, and DAMs (differentially abundant metabolites) were filtered using a combination of the *p*-value of Student’s *t*-tests with the VIP (variable importance of the projection) score of the OPLS-DA model. The metabolites were considered different when the *p*-value of the *t*-test < 0.05 and VIP ≥ 1.

### 2.11. Statistical Analysis

Means, standard errors, and significance analysis of differences were calculated by SPSS 22.0 software. Tukey’s multiple comparison test was used to determine the significance of differences between samples. The determination of physical and chemical indexes of samples and transcriptome sequencing carried out at least 3 biological repetitions, while metabolic group determination used 6 biological repetitions. The figures in this paper were drawn by originPro 8.0 (OriginLab Corporation, Northampton, MA, USA) and Photoshop C5 software (Adobe Systems Inc., San Francisco, CA, USA).

## 3. Results

### 3.1. Changes of Sucrose, Fructose, Glucose, and Maltose

The contents of sucrose, fructose, glucose, and maltose in the three sweetpotato cultivars were determined by HPLC (Figure 2). The sucrose content was changed between 90 and 130 days, especially in X27 under different temperature treatments. However, under different temperature treatments, the fructose content of the three sweetpotato cultivars showed significant changes in the two growth stages, but this change had no obvious regularity. The glucose content of the two high saccharification cultivars Y25 and Z13 was lower than that of the low saccharification cultivar X27, especially at the growth stage S2. Therefore, it can be indicated that glucose may contribute to the final sweetness of sweetpotato. The maltose content of the three cultivars gradually increased with the increase of temperature. It was higher in the two high saccharified cultivars than low saccharified cultivar X27 between 90 and 130 days. From the results, it was found that the maltose content increased nearly 10 times before and after temperature treatment, indicating that it made an important contribution to the final sweetness of sweetpotato.

### 3.2. Differences of Starch Content, Amylose Content, and Starch Gelatinization Temperature

The starch content of raw and steamed sweetpotato storage roots was determined by starch content determination kit. The results showed that there was no significant difference in starch content of raw storage roots in the S1 period. During the S2 period, it was significantly higher in raw storage roots of cultivars X27 and Z13 than cultivar Y25. After steaming, the starch content significantly decreased in Y25 and Z13 only, but X27 did not show any change in both S1 and S2 stages (Figure 3A). By comparing the changes of starch content before and after cooking, it was found that the starch content in Y25 and Z13 were significantly higher than X27 (Figure 3B). The gelatinization temperature reflected the characteristics of starch and affected its enzymatic hydrolysis. In two growth stages, the starch gelatinization temperature of Y25 and Z13 was significantly lower than that of X27, which was determined by RVA (Figure 3C). The amylose content of sweetpotato cultivar Z13 was significantly higher than that of the other two cultivars in the two growth stages (Figure 3D). In summary, it could be seen that there are great differences in starch physical and chemical properties between the high (Y25 and Z13) and low (X27) saccharification sweetpotato cultivars.

### 3.3. Changes of Total Amylase Activity

The changes of total amylase in different sweetpotato cultivars under different temperature treatments were determined by enzyme-linked immunosorbent assay (Elisa). The results showed that the total amylase activity increased in the beginning, then decreased with the increase of temperature, especially in S2 period. This revealed that the activity of total amylase was affected by the change of temperature. The activity of amylase decreased when the optimum temperature of amylase activity was reached (Figure 4).

### 3.4. Taste Evaluation

The comprehensive scores of Y25 and Z13 which included viscosity, sweetness, and fiber content were significantly higher than those of X27 (Figure 5A). Similarly, the sweetness scores of Y25 and Z13 were significantly higher than those of X27 (Figure 5B).

### 3.5. Transcriptome Analysis

The storage root samples of three sweetpotato cultivars treated at 60 °C and CK were sequenced for three times by transcriptome sequencing independently, in order to correctly and objectively reflect the molecular mechanism of sweetpotato saccharification at 60 °C. The total number of bases was greater than 6 GB and an average of 6,618,347,760 clean data was obtained from different samples after filtering. The Q30 percentage and GC content were over 92% and 48%, respectively (Appendix A). The mapped sequence was greater than 77%, including about 70% of the unique mapped reads and about 5% of the multiple mapped reads (Appendix A).

### 3.6. Identification of DEGs and Analysis

Depending on the previous analysis, the sweetpotato cultivars Y25 and Z13 were taken as the high saccharification group, while the cultivar X27 was in the low saccharification group. The screening of differential genes mainly included the genes with different expression in cultivars Y25 and Z13 and those with no differential expression of X27, and the genes whose expression patterns of Y25 and Z13 are opposite to those of X27 in sweetpotato cultivars (the red area in the Figure 6A).

A total of 1918 DEGs were obtained during the S1 growth stage, including 439 up-regulated genes and 1469 down-regulated genes in Y25, while Z13 included 454 up-regulated genes and 1454 down-regulated genes, as well as 10 genes with inconsistent expression trends. During the growth period of S2, a total of 1520 DEGs were screened, including 392 up-regulated and 1096 down-regulated genes in Y25, and 409 up-regulated and 1079 down-regulated genes in Z13, also including 32 DEGs with opposite trends (Figure 6B).

To verify the reliability of transcriptome data, nine EDGs were randomly selected for qRT-PCR verification. The results verified that transcriptome data and qRT-PCR data had the same trend of Illumina High throughput sequencing (Appendix A). Correlation analysis confirmed that the correlation coefficient was 0.7995 (R^2^ = 0.7995), which proved that the transcriptome data were reliable (Figure 6C).

According to the principal component analysis of the sample relationship, the sweetpotato samples had a good degree of separation before and after treatment, indicating that the experimental treatment was effective and feasible (Figure 6D).

### 3.7. Enrichment of GO Terms and KEGG Pathway Analysis

The gene ontology (GO) term was assigned to the annotated DEGs identified in this experiment. A total of 1918 and 1520 DEGs from S1 and S2 growth stages, respectively, were classified into three categories including biological process, cellular component, and molecular function. After the distribution of DEGs on the S1 and S2 growth stages, the terms of the different categories and quantitative proportions were similar. The number of DEGs which classified into metabolic process, cellular process, cell, cell part, catalytic activity, and binding term accounted for a large proportion (Figure 7A,B). The *p*-value was used to further screen the GO terms with significant differences. When *p* < 0.05 the GO terms were considered significantly different. Between S1 and S2 growth stages, 80 and 57 GO terms with significant differences were obtained, respectively. The GO terms relating to saccharification in S1 growth stages including “Sucrose metabolic process” (GO:0005985), “Response to sucrose” (GO:0009744), “Response to carbohydrate” (GO:0009744), “Hydrolase activity” (GO:0004553, GO:0016798), and “Amylase activity” (GO:0016160) were purposefully selected. In the same way, the significant differences obtained in S2 were completely different from that of S1, which mainly included “Maltose metabolism process” (GO:0000023), “Carbohydrate metabolism” (GO:0005975), “Hexose metabolism” (GO:0019318), and “Hydrolase activity” (GO:004553, GO:0016811) (Appendix A).

The specific metabolic pathways in the regulation of sweetpotato storage root saccharification at 60 °C were excavated by KEGG enrichment analysis. According to the analysis of the metabolic pathways co-enriched in the two growth stages, there were five common pathways in the first 20 enrichment pathways, which were “Starch and sucrose metabolism” (ko00500), “Biosynthesis of secondary metabolites” (ko01110), Plant hormone signal transduction (ko04075), “Galactose metabolism” (ko00052), “Isoquinoline alkaloid biosynthesis” (ko00950), and “Brassinosteroid biosynthesis” (ko00905) (Figure 8A,B). The metabolic pathways related to saccharification were mainly starch and sucrose metabolic which enriched 31 and 17 DEGs in S1 and S2 growth period, respectively (Appendix A). Further annotation and statistics of the enriched metabolic pathways showed that there were 83 and 59 annotations related to carbohydrate metabolism in 90 d and 130 d, separately (Figure 8).

### 3.8. Statistics Analysis of DAMs

Screening DAMs by constructing orthogonal partial least squares model (Appendix A) which was verified by using permutation test, the metabolites were considered different when the *p*-value of the *t*-test < 0.05 and VIP ≥ 1. The DAMs were analyzed by principal component analysis to observe the repeatability and separation of the samples. The results showed that the repeatability and separation of the samples before and after treatment were satisfactory, indicating that the experimental treatment was effective (Appendix A).

### 3.9. KEGG Annotation and Enrichment Analysis of DAMs

Six biological replicates of each sweetpotato samples were analyzed to identify DAMs. According to the difference between positive and negative channel the number of DAMs of each sample was between 1014 and 1171. In S1 stage, the number of down-regulated DAMs of sweetpotato cultivar X27 was significantly higher than that of up-regulated, while other cultivars showed the opposite trend in S1 and S2 stages. The number of DAMs of sweetpotato cultivar X27 was the highest as compared to other cultivars (Figure 9A,B). KEGG pathway-enrichment analysis of the DAMs was then carried out.

A total of five KEGG categories were significantly enriched, including “metabolism”, “Genetic information processing”, “Environmental information processing”, “Cellular processing”, and “Human disease”. The terms related to “metabolism” (“global and overview”, “amino acid metabolism” and “Carbohydrate metabolism”) were significantly enriched. There were 162 annotations enriched in “Carbohydrate metabolism” which was a major concern. The other four categories contained only a small number of KEGG pathway annotations (Figure 9C). Furthermore, the first 20 terms enriched by KEGG were further analyzed in samples of three sweetpotato cultivars in S1 and S2 stages (Appendix A). The results showed that during the S1 growth stage, seven metabolic pathways were co-enriched into three groups of samples, including “ABC transporters”, “beta-Alanine metabolism”, “Aminoacyl-tRNA biosynthesis”, “Starch and sucrose metabolism”, “2-Oxocarboxylic acid metabolism”, “Porphyrin and chlorophyll metabolism”, and “Histidine metabolism”. Similarly, during the S2 growth period, four metabolic pathways were co-enriched into three samples which covered “ABC transporters”, “Starch and sucrose metabolism”, “Cyanoamino acid metabolism”, and “Arginine and proline metabolism”. In these enriched pathways, “Starch and this sucrose metabolic” pathway was co-enriched in S1 and S2, and was also enriched and given special attention in the transcriptome data analysis, which demonstrated that it was the vital metabolic pathway of sweetpotato saccharification at high temperature. Interestingly, “ABC transporters” pathway was also co-enriched in S1 and S2, which might play an important role in sweetpotato saccharification (Table 1).

### 3.10. Correlation Analysis of DEGs and DAMs

The association trend between DEGs and DAMs was analyzed using the O2PLS model. O2PLS is an unsupervised model, which could objectively describe whether there was a trend of association between two data groups [30]. The O2PLS model was used to predict the first 25 DEGs and DAMs with a large contribution. In S1 stage, two DEGs (G489601 and G38875) related to carbohydrate metabolism were predicted in 25 members. In addition, 25 members also were included in zinc finger structural proteins (G5912 and G12636), ubiquitin (G45167), and stress resistance proteins (G48119, MSTRG.5799, and MSTRG.69786). Similarly, in S2 stage, among the 25 differential genes there were two genes related to carbohydrate metabolism (G45445 and MSTRG.45913), ubiquitin protein (G30084), threonine-protein kinase (G17264), and aquaporin (G286). However, 25 DAMs had no functional annotations and their material properties could not be known (Figure 10).

In order to mine the more valuable information related to sweetpotato saccharification in the data, correlation analysis of DEGs and DAMs was performed (Figure 11). The darker the red or blue shown in the figure, the stronger the correlation between DEGs and DAMs, and the more significant. In contrast, the lighter color areas indicated that the correlation between DEGs and DAMs was much weaker. The DEGs related to sugar were further screened and the correlation between them and differential metabolites was observed. In S1 period, a total of nine genes (G24806, G38211, G6838, G25158, G33024, G6920, G36626, G41176, and G38871) were selected. These nine genes also showed a strong correlation with NEG02758, NEG07822, NEG00118, NEG07285, POS01976, NEG 05968, POS00238, NEG03333, NEG03447, NEG03214, POS04841, POS04913, NEG00226, and other DAMs. At the same time, there were seven DEGs, related to glucose metabolism obtained during S2 which had a strong correlation with many differential metabolites including POS00722, POS04603, NEG01587, NEG01599, POS00717, POS02627, NEG03205, POS02602, POS02631, NEG01512, NEG01503, POS02599, POS00060, POS04850, and POS10906 (Appendix A). Thus, it is suggested that DEGs and DAMs with strong correlation could be selected for further study of sweetpotato saccharification.

### 3.11. Enrichment Pathway Analysis of DEGs and DAMs

The expression levels of DEGs and DAMs were co-annotated into the starch and sucrose metabolic pathways simultaneously (Figure 12). Eighteen functional genes were annotated, and each functional gene contained multiple unigenes. These functional genes include genes related to starch synthesis and degradation (such as GBSS (granular starch synthase), SS (starch synthase), GBE (branching enzyme), AMY (α-amylase)), sucrose synthesis, and degradation related genes (such as SuS (sucrose synthase), SPS (sucrose phosphate synthase), INV (sucrose invertase)), trehalose related genes (such as TPS (trehalose phosphate synthase), TPP (trehalose phosphorylase)), and glucose related genes (α-GLU, β-GLU (glucosidase), HK (fructose kinase), GPI (Glucose-6-phosphate isomerase), etc.). The expression of most genes showed a significant down-regulation, however there might be differences in different unigenes aligned to the same gene including SuS, SPS, HK, SS, GBSS, and GBE. We would select some of these saccharification-related genes for functional verification in the future. Moreover, a total of seven DAMs were enriched in the pathway, i.e., sucrose (NEG00034), D-fructose-6P (POS02377, POS03021), D-glucose-6p (POS00383), levan (POS00114), β-D-glucose (POS01398), D-xylose (NEG00106) and trehalose (NEG00042). The results showed that trehalose was up-regulated in three sweetpotato cultivars during S1 and S2 periods. However, β-D-glucose showed a down-regulation in the three sweetpotato cultivars during S1 period and up-regulation during S2 period. Furthermore, there were no significant changes of other substances among the three sweetpotato cultivars at the different periods.

## 4. Discussion

### 4.1. Experimental Conception and Design

This experiment was conducted to clarify whether samples treated at 60 °C could objectively and truly reflect the real situation of sweetpotato storage root saccharification at high temperature. RNA translation into protein had a certain lag, and at normal temperature, RNA could be translated into proteins to perform specific functions. In the current results, when the sweetpotato storage roots were subjected to 60 °C, a large amount of RNA was induced and translated into corresponding proteins stored in the storage roots, but the induced enzymes could not decompose the ungelatinized starch granules and no maltose was produced. When the temperature rises to 80 °C, RNA decomposed, starch grains began to gelatinize, and the induced enzyme reacted with gelatinized starch to produce a large amount of maltose. Therefore, it could be found that the induced RNA and protein at 60 °C could represent the real situation of the whole high temperature saccharification in a certain sense (Appendix A).

### 4.2. Changes of Genes and Metabolites Related to Sucrose Metabolism in Sweetpotato during High Temperature Saccharification

According to the changes of sweetpotato sugar components determined by HPLC, it was found that the content of maltose increased sharply with the increase of treatment temperature, while sucrose, glucose, and fructose changed little or no significant regularity. Shen et al. obtained the same results by studying the soluble sugar components of sweetpotato storage roots, which showed that maltose contributed the most to the sweetness of sweetpotato after steaming [31]. Not surprisingly, through the enrichment analysis of DEGs and DAMs, starch and sucrose metabolic pathways were enriched, which included the maltose synthesis pathway. Many DEGs in this pathway were screened, including SuS, SPS, HK, SS, GBSS, GBE, AMY, and so on. Although β-amylase proved to be related to the synthesis of maltose in sweetpotato storage roots [9,32], the identified DEGs were also involved in the synthesis of maltose. SS, GBSS, and GBE might affect the gelatinization of starch by regulating the structure of starch, thus regulating the transformation of starch to maltose [18]. Sucrose, D-fructose-6P, D-glucose-6p, levan, β-D-glucose, trehalose, and other metabolites were identified to participate in the saccharification process of sweetpotato. Through the analysis of the results, we could clearly find that the pathways involved in sweetpotato saccharification was starch and sucrose metabolic pathways.

### 4.3. Transcription Factor Families (TF Families) and Inhibitors Related to Saccharification

The current results showed that a main series of TF families were obtained by differential expression screening (Table 2). Previous studies have found that MYB proteins were key factors in regulatory networks controlling development, metabolism, and responses to biotic and abiotic stresses [33]. Similarly, in rice, it has been proven that MYB can regulate the activity of amylase through sugars and hormones [34]. In this experiment, several differentially expressed MYB TFs were also identified, among which some TFs with large multiples of differences, i.e., G12956, G33975, and G36436, were involved in the saccharification metabolism of sweetpotato. Furthermore, WRKY TF families were also confirmed to be involved in sugars metabolism and signal transduction in plants [35,36]. Analysis of NDPK3a promoter identified two boxes that have previously been reported to be involved in sugar signaling in barley by binding to SUSIBA2 protein which belongs to WRKY TF showing a possible involvement of WRKY TFs in sugar induction of NDPK3a [37]. SPF1 identified WRKY protein in sweetpotato, which can bind to SP8 elements upstream of the sugar-responsive sporamin and β-amylase genes whose expression was induced by sugar [38]. In addition, in barley aleurone cells, another WRKY factor HvWRKY38 inhibited GA-induced Amy32b expression by binding to Amy32b promoter [39]. Some WRKY TF family members (e.g., G45220, G11050, and G15453) were also obtained from the data of this study, which can provide genetic resources for further research related to sweetpotato saccharification.

In the process of plant metabolism, complex metabolic pathways were involved, including not only the functions of key genes, but also a series of transcriptional factors and inhibitors that play a regulatory role. Amylase, especially α-amylase and β-amylase, play an important role in starch and sucrose metabolic pathways which was also enriched by transcriptome and metabolic analysis in this study (Appendix A). Similarly, the function of amylase was also regulated by some inhibitors reported in several species. Most amylase inhibitors inhibit amylase by forming complexes, thus blocking the active site or changing the conformation of the enzyme, resulting in the decrease of its catalytic activity [40]. Amylase inhibitors have been confirmed in potatoes, coffee, and beans, and could affect amylase activity [41,42,43]. No amylase inhibitors were found in transcriptome data of this study, but some other inhibitors were found, including pectin-esterase inhibitor (G5479, G42776), Serine-type endopeptidase inhibitor (MSTRG.35397), and proteinase inhibitor (G28472).

### 4.4. Network Analysis of DEGs and DAMs

For further understanding of the relationship between DEGs and DAMs, a network diagram of them was constructed (Appendix A). Network relationship analysis can effectively analyze the association between multiple gene clusters and multiple metabolites [44]. The current results show that there are some important metabolites associated with many DEGs in both S1 and S2 stages, including NEG02578, NEG03095, POS02873, NEG03205, NEG01503, and POS04850. However, some of the metabolites lacked annotations, and we do not know what kind of substance they are. We can only roughly speculate about the metabolic pathway that these metabolites belong to through genetic annotations associated with them. For example, the metabolite NEG01503 and its associated genes G25936, G39341, and G619, were related to surge metabolism, and it could be considered that the metabolite NEG01503 might be related to glucose metabolism. However, there were also some problems in this conjecture that some transcription factors, inhibitors, and regulatory genes might jointly regulate the synthesis of a substance, but there was no upstream and downstream regulatory relationship between them, which makes it difficult to speculate the participation pathway of the substance. Therefore, more accurate experimental verification will be needed if we want to accurately understand the properties of metabolites.

## 5. Conclusions

In the present study, sweetpotato samples in the process of saccharification at high temperature were assayed by the transcriptome and metabolic, and a total of 1918 and 1520 DEGs and 230 and 192 DAMs were identified from the samples during S1 and S2, respectively. Functional annotation and analysis of DEGs showed that some functional genes were enriched in starch and sucrose metabolic pathways, and some transcription factors (such as MYB family, WAKY family, bHLH family, etc.) and inhibitors (pectinesterase inhibitor, Serine-type endopeptidase inhibitor) were screened out, which were likely to be involved in regulating the saccharification process of sweetpotato. In addition, the DAMs obtained by OPLS-DA model analysis were also enriched on the starch and sucrose metabolic pathway, indicating that this pathway was a vitally important pathway for saccharification of sweetpotato. The current transcriptome sequencing data of sweetpotato under the condition of high temperature saccharification can provide a valuable resource for table use sweetpotato breeding research and offer novel insights into sweetpotato saccharification. In addition, it also supplies new candidate genes, which can be used to guide future research on the breeding of new sweetpotato cultivars with high saccharification.

## Figures and Tables

**Figure 1 ijms-22-06641-f001:**
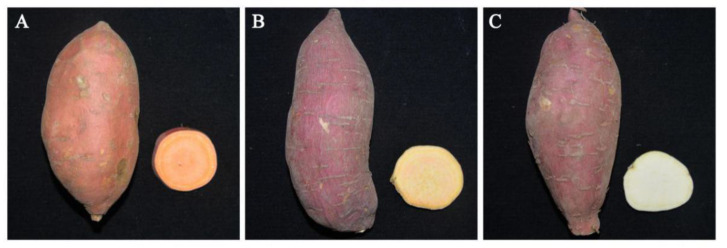
Phenotypic characteristics of 3 sweetpotato cultivars that represent the storage roots and the storage root cross-sections of (**A**) Y25, (**B**) Z13, and (**C**) X27, respectively.

**Figure 2 ijms-22-06641-f002:**
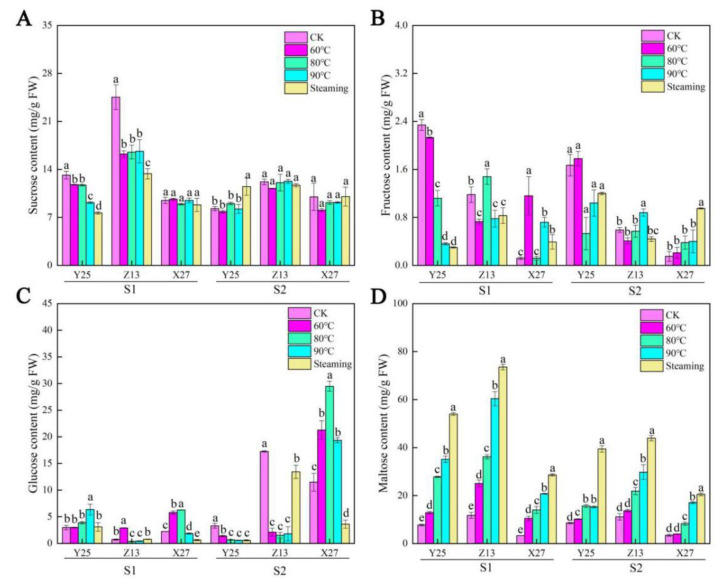
(**A**) Sucrose, (**B**) fructose, (**C**) glucose, and (**D**) maltose contents in three sweetpotato cultivars under different temperature treatments. The changes of sucrose, fruc-tose, glucose, and maltose content in three sweetpotato cultivars under different temperature treatments, respectively. Bars not sharing a common letter differ (*p* < 0.05) by one-way ANOVA and Tukey’s multiple comparison test.

**Figure 3 ijms-22-06641-f003:**
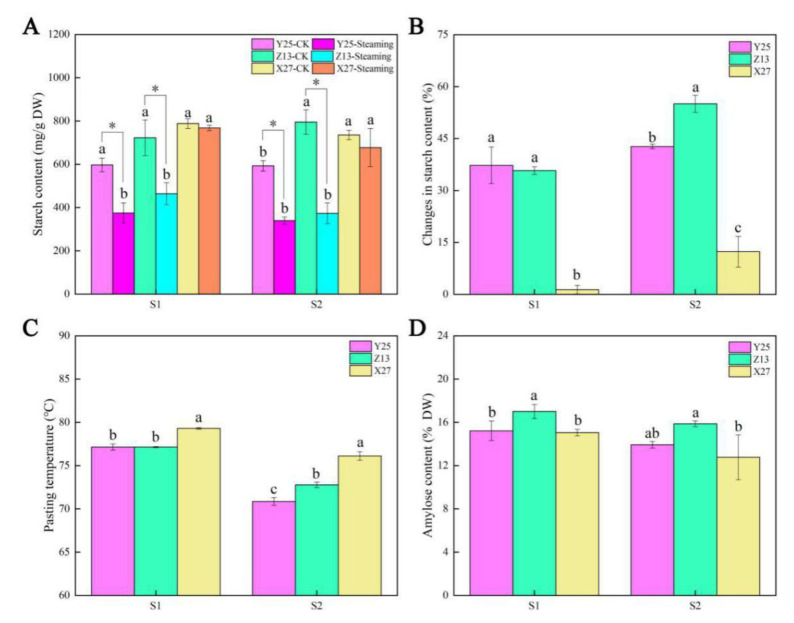
(**A**) Starch content, amylopectin content, and gelatinization temperature of the samples at 90 d and 130 d; (**B**) The changes of starch content in sweetpotato before and after steaming; (**C**) Changes in gelatinization temperature of sweetpotato starch; (**D**) Amylose content of sweetpotato. The SE of three biological repetitions is shown as an error bar. Bars not sharing a common letter differ (*p* < 0.05) by one-way ANOVA and Tukey’s multiple comparison test. Asterisks indicate statistical differences between CK and steaming values (*p* < 0.05, Student’s t-test).

**Figure 4 ijms-22-06641-f004:**
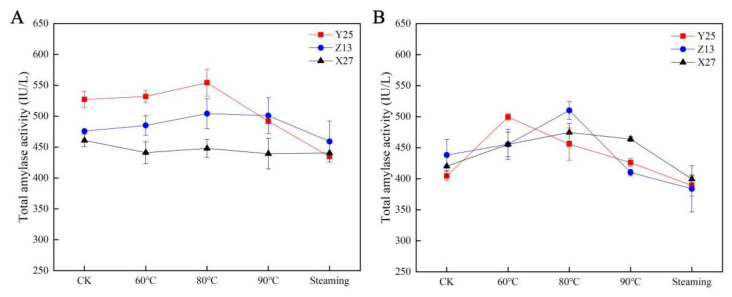
Total amylase activity of sweetpotato in two growth stages. Total amylase activity in sweetpotato between (**A**) 90 and (**B**) 130 days, respectively. The SE of three biological repetitions is shown as an error bar.

**Figure 5 ijms-22-06641-f005:**
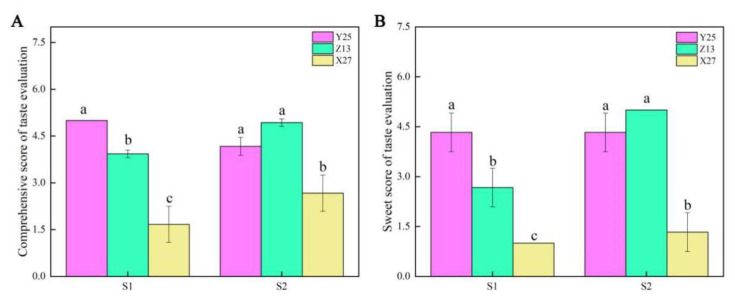
Comprehensive score and sweetness score of three sweetpotato cultivars in two growth stages. (**A**) The changes of comprehensive scores in taste evaluation of three sweetpotato cultivars; (**B**) The changes of sweet scores in taste evaluation of three sweetpotato cultivars. The SE of three biological repetitions is shown as an error bar. Bars not sharing a common letter differ (*p* < 0.05) by one-way ANOVA and Tukey’s multiple comparison test.

**Figure 6 ijms-22-06641-f006:**
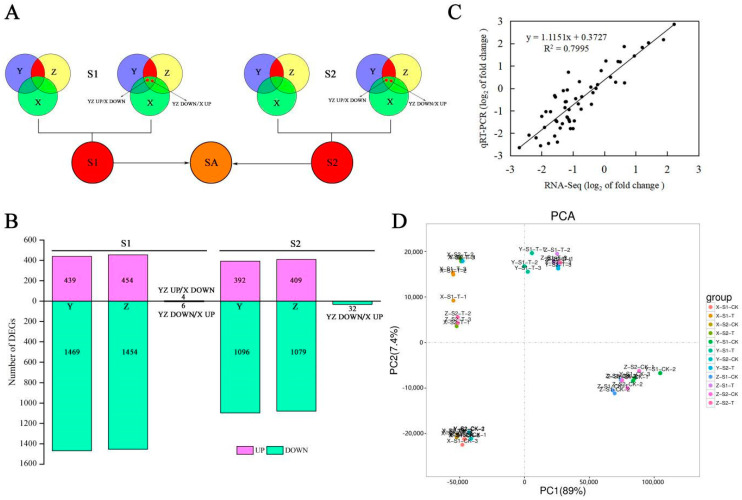
Statistics and verification of DEGs in transcriptome and principal component analysis (PCA) of samples. (**A**) Analysis strategy of transcriptome data. (**B**) Statistics of the number of DEGs in two growth stages. (**C**) Correlation analysis of transcriptome and qRT-PCR data. (**D**) PCA of transcriptome samples.

**Figure 7 ijms-22-06641-f007:**
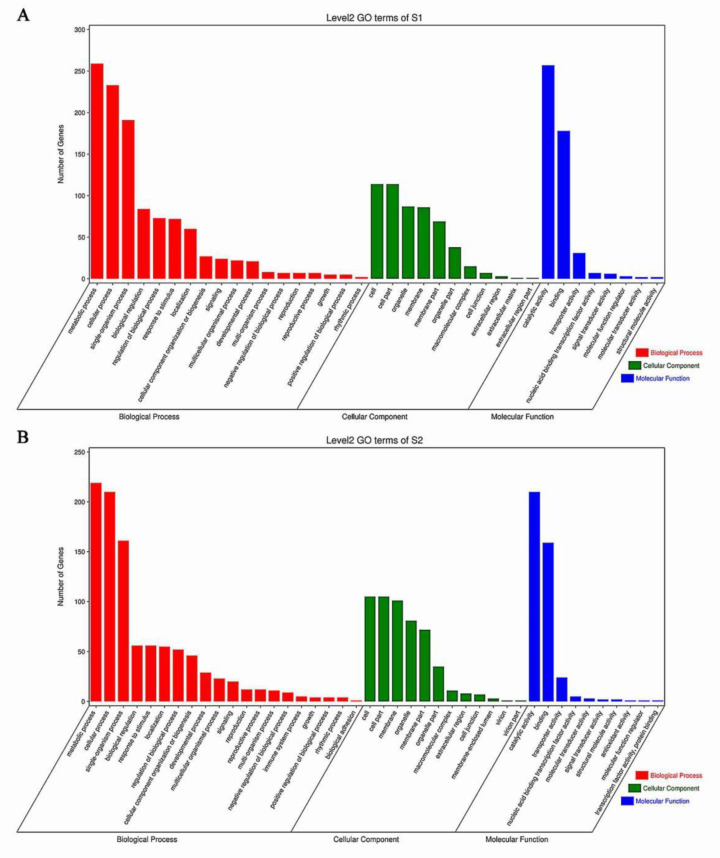
GO annotation analysis of transcriptome. Statistics of the number of DEGs in 3 function categories of GO annotation in sweetpotato for (**A**) 90 d and (**B**) 130 d, respectively. The red, blue, and green columns represent the terms of biological process, cellular component, and molecular function, respectively.

**Figure 8 ijms-22-06641-f008:**
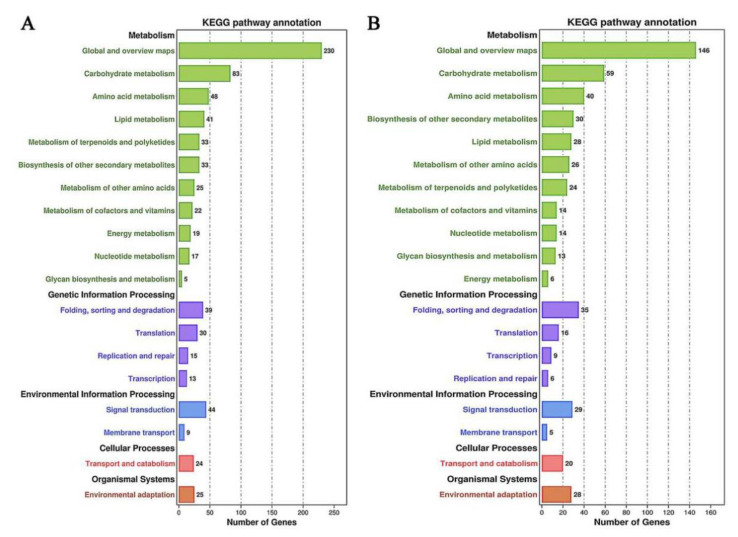
KEGG pathway analysis of transcriptome. The statistics of the number of DEGs in different metabolic pathways were obtained by KEGG enrichment analysis in sweetpotato for (**A**) 90 d and (**B**) 130 d, respectively. The number marked on the column was the number of DEGs enriched in this metabolic pathway.

**Figure 9 ijms-22-06641-f009:**
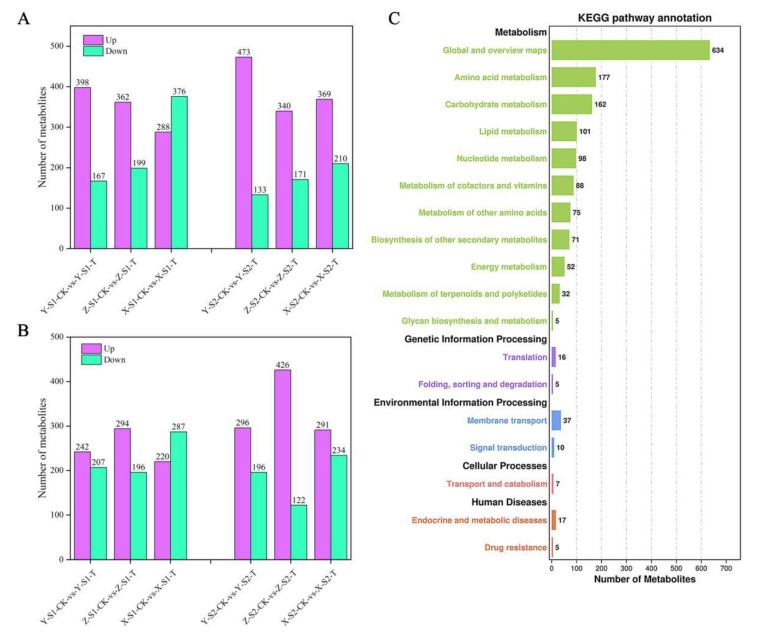
Quantitative statistics of DAMs and their enrichment in different metabolic pathways. (**A**,**B**)The number of DAMs in positive and negative channel, respectively. The six columns on the left are the statistical results of S1 DAMs in the S1 growth period, and the six columns on the right are the statistical results of S2 growth period. The purple column is the number of metabolites with up-regulated abundance, and the green column is the statistics of the number of metabolites with down-regulated abundance; (**C**) The number of DAMs in different metabolic pathways by KEEG enrichment analysis. The number marked on the column was the number of DAMs enriched in this metabolic pathway.

**Figure 10 ijms-22-06641-f010:**
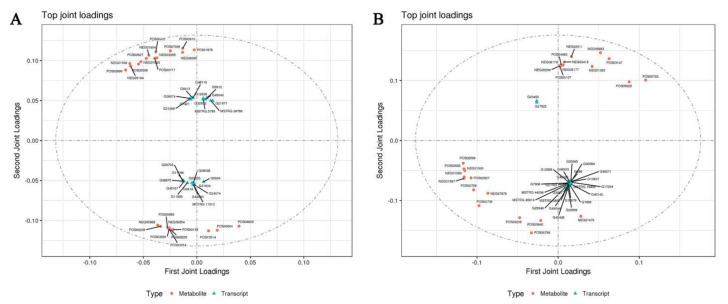
Correlation analysis of DEGs and DAMs. Correlation analysis of DEGs and DAMs in (**A**) S1 and (**B**) S2, respectively. In the picture, the orange dots represent DAMs, and the blue triangles represent DEGs. The top 25 DEGs and DAMs with high correlation were marked in the figure.

**Figure 11 ijms-22-06641-f011:**
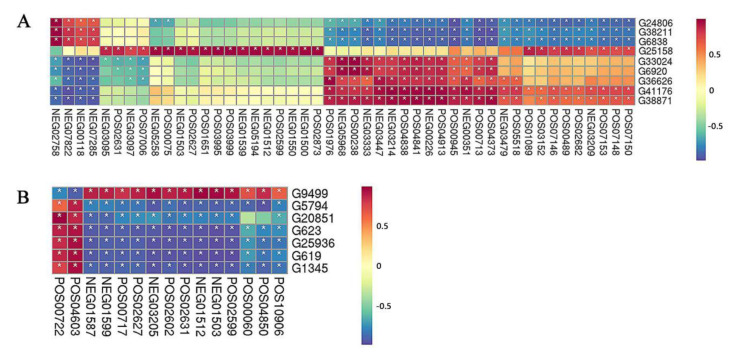
Correlation analysis of DEGs and DAMs. The analysis results of sweetpotato at the growth stages (**A**) S1 and (**B**) S2, respectively. The * marked in the figure represented a significant correlation between DEGs and DAMs. The redder the color, the more significant the positive correlation and the bluer the color, the more significant the negative correlation.

**Figure 12 ijms-22-06641-f012:**
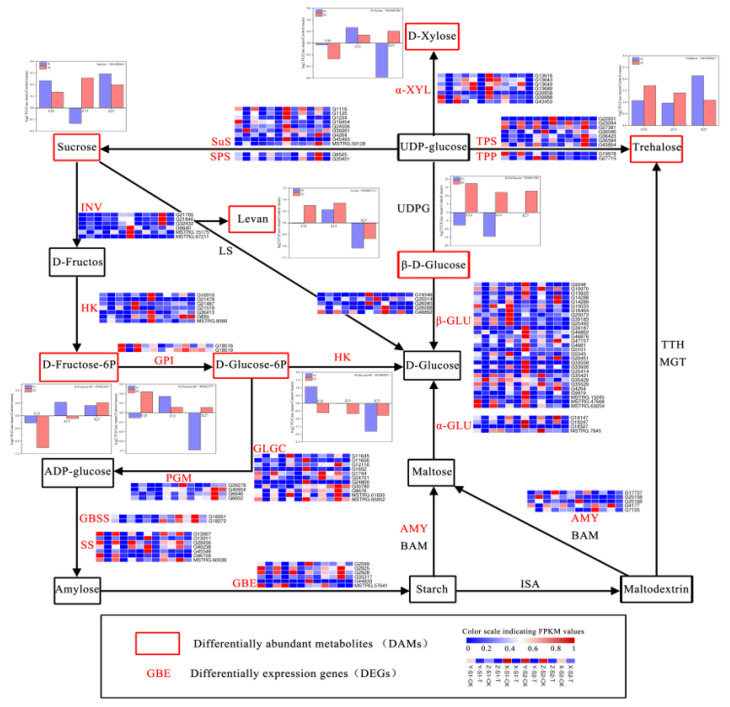
Starch and sucrose metabolic pathways which enriched by DEGs and DAMs. The red font marked gene in the picture represents the DEGs enriched by the transcriptome, and the heat map next to them shows the expression of three sweetpotato varieties in S1 and S2. The metabolites marked in the red box represented the DAMs enriched by the metabolites determined by the metabolome, and the bar chart next to them shows the multiple differences between the treated and CK samples of three sweetpotato varieties in S1 and S2 periods.

**Table 1 ijms-22-06641-t001:** Number of DAMs in co-enriched metabolic pathway between S1 and S2 growth periods.

Terms	S1					
	Number of DAMs			*q* Value		
	Y25	Z13	X27	Y25	Z13	X27
ABC transporters	10	7	9	0.0060	0.1358	0.0266
beita-Alanine metabolism	2	2	2	0.8594	0.9314	0.7337
Aminoacyl-tRNA biosynthesis	4	2	4	0.4053	0.9314	0.3486
Starch and sucrose metabolism	1	1	2	0.8594	0.9314	0.5925
2-Oxocarboxylic acid metabolism	4	3	5	0.8594	0.9314	0.6510
Porphyrin and chlorophyll metabolism	3	1	3	0.4053	0.9314	0.3486
Histidine metabolism	3	2	3	0.8594	0.9314	0.7337
Terms	S2					
ABC transporters	4	5	8	0.7417	0.6227	0.2009
Starch and sucrose metabolism	2	1	2	0.3624	0.8954	0.7974
Cyanoamino acid metabolism	1	2	2	0.8803	0.6227	0.7974
Arginine and proline metabolism	5	3	4	0.2321	0.8884	0.7974

**Table 2 ijms-22-06641-t002:** Differentially expressed transcription factors Log2 (fc) represented the multiple of absolute difference between the sample treated at 60 °C and CK.

S1					S2				
Types	ID	Log2 (fc)			Types	ID	Log2 (fc)		
		Y25	Z13	X27			Y25	Z13	X27
MYB	G12956	−4.63	−1.21	−0.68	MYB	G31499	7.46	3.53	2.09
	G31499	−4.51	1.91	0.78		G3991	−4.32	−3.93	−2.68
	G33975	−4.21	−5.26	−1.38		G18652	−3.19	2.15	−0.92
	G26043	−3.45	−2.17	−1.70		G18587	−2.98	−3.10	−1.55
	G16218	−2.99	−3.19	−1.62		G36436	−2.92	2.00	−1.32
	MSTRG.35719	−2.42	−2.50	−2.47		G18679	−2.35	−2.28	−1.74
	G4815	−1.95	3.89	−3.59		G34548	−2.23	−1.52	−1.00
	G34342	1.37	1.99	0.95					
	MSTRG.9428	1.50	1.34	0.89	HSF	G21417	−3.52	−2.80	−0.50
	G30675	2.19	1.98	0.80					
	G36436	2.67	1.32	−0.42	ERF	G27649	−5.31	−8.96	−0.43
						MSTRG.50858	−3.09	−1.58	−0.87
HSF	G38291	−10.37	−2.31	−4.74		G42381	1.52	2.35	0.77
	G21545	−3.19	−1.19	−0.22					
	G31518	−2.56	−1.81	−1.65	bHLH	G21240	−6.45	−6.16	−1.85
	G23088	−2.21	−2.02	−2.21		G12959	−5.03	−2.54	−2.78
	G3345	−1.84	−0.41	−2.67		G46039	−4.92	−4.56	−2.77
	G25273	1.83	1.38	0.68		G2655	−4.50	−10.36	9.15
	G18195	1.90	1.60	−0.43		G27729	−4.36	−3.89	−0.51
						G29441	−3.52	−3.09	−1.60
ERF	G23031	−11.73	−4.86	−3.35		G1375	−3.46	−1.46	−1.55
	G28150	−4.72	−4.38	−8.23		G15876	−2.65	−1.20	−1.35
	G32004	−4.37	−3.65	−3.21		G29812	−2.58	−3.09	−2.03
						MSTRG.47017	−2.38	−2.92	−1.65
bHLH	G6860	−9.73	−9.61	−1.51		G12549	−2.13	−2.03	−0.85
	G16325	−5.74	−3.65	−2.34		G42307	−2.07	−1.97	−1.40
	G17961	−4.53	−2.97	−1.30		G19671	−1.98	−3.58	−1.97
	G25250	−2.99	−3.28	−2.20		G16702	−1.87	−1.26	−1.44
	G19671	−2.18	−1.93	−1.58		G16207	−1.76	−1.87	−1.49
	G31307	3.32	2.23	0.63					
					WRKY	G11050	−10.12	−9.37	−8.13
WRKY	MSTRG.69878	−3.38	−2.42	−2.38		G15453	−3.26	−3.66	−1.01
	G7445	2.95	2.35	0.91		G41003	−2.88	−2.17	−1.24
	G45220	4.68	3.27	−0.73		G7665	−2.67	−3.23	−2.09

## Data Availability

RNA-seq data will soon be submitted to the NCBI Sequence Read Archive (SRA). Additional data supporting the findings are included in the article.

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
