# Peer review of "Transcriptomic and Metabolic Profiling of High-Temperature Treated Storage Roots Reveals the Mechanism of Saccharification in Sweetpotato (Ipomoea batatas (L.) Lam.)"

_ijms, 2021, doi:10.3390/ijms22136641_

Round 1

Reviewer 1 Report

“Yanshu 102 25 (Y25), Zheshu 13 (Z13) and Xushu 27 (X27)”; Theauthors should clarify the botanical names (proper nomenclature).

Author Response

  1. In the introduction, we have added and modified some contents appropriately, such asline 61-62,line 78.
  2. For the description of the experimental design and method, in order to make readers better understand, we have made appropriate revision according, such as line 101-102,line 117-118,line 122-123, line152-153, etc..
  3. For the questions raised by expert, “Yanshu 25 (Y25), Zheshu 13 (Z13) and Xushu 27 (X27); The authors should clarify the botanical names (proper nomenclature).” We made changes in the text, changing Yanshu25 (Y25), Zheshu13 (Z13) and Xushu27 (X27) to Y25/Y (Ipomoea batatas(L.) Lam. cv 'Yanshu25'), Z13/Z (Ipomoea batatas (L.) Lam .cv 'Zheshu13') and X27/X (Ipomoea batatas (L.) Lam. cv 'Xushu27').

Reviewer 2 Report

This is an very interesting scientific work with high value.

The version need some revision (M&M, Captions of Figs, Scientific writing style).

Remarks/Comments: see file.

Author Response

  1. The version need some revision (M&M, Captions of Figs, Scientific writingstyle).

According to the requirements and suggestions put forward by the reviewers, we have revised the spelling, sentences, figs and instructions of the whole article, which have been marked in red. At the same time, the language style has been properly modified and the ambiguous sentences and words which will bring ambiguity to the reader have been explained or deleted.

Reviewer 3 Report

In this study, the molecular mechanism of saccharification of sweetpotatoes by heat treatment is investigated. The experiment is well done and the authors present very interesting data. Therefore, I think that it is a suitable work to be published in IJMS

I have a question. It seems likely that the low saccharification of cultivar X27 is due to the suppressed degradation of starch. However, the amylase activity of cultivar X27 was not much lower than those of cultivars Y25 and Z13. The molecular mechanism of the low sugar content of cultivar X27 is unknown. The authors should discuss this mechanism in the discussion section.

Minor point

Page 1, "Correspondence" is dubbed.

Page 1, I think that a list of abbrevations is not required.

Author Response

  1. For the questions raised by the reviewers, "It seems likely that the low saccharification of cultivar X27 is due to the suppressed degradation of starch. However, the amylase activity of cultivar X27 was not much lower than those of cultivars Y25 and Z13. The molecular mechanism of the low sugar content of cultivar X27 is unknown. The authors should discuss this mechanism in the discussion section."

According to the research background, we know that the sweetness of sweetpotato is not only positively correlated with amylase, but also related to the gelatinization properties of sweetpotato starch. Although there was no significant difference in amylase activity and amylopectin content between X27 and Y25/ Z13, the gelatinization characteristics of starch were significantly different, especially the gelatinization temperature. Some reports showed that there was a significant negative correlation between gelatinization temperature and maltose content of sweetpotato starch (reference 9/15). Therefore, we infer that the difference in sweetness between X27 and Y25/Z13 is mainly due to the difference of starch gelatinization properties.

  1. Minor point

Page 1, "Correspondence" is dubbed.

We have deleted the extra words "Correspondence". line12

Page 1, I think that a list of abbreviations is not required.

According to the advice of reviewers, we have deleted the abbreviations at the beginning of the article, mainly because the key abbreviations have been explained in the article.